# Effects of Hygrothermal Aging and Cyclic Compressive Loading on the Mechanical and Electrical Properties of Conductive Composites

**DOI:** 10.3390/polym14235089

**Published:** 2022-11-23

**Authors:** Shuwang Yi, Long Xie, Zhi Wu, Weiming Ning, Jianke Du, Minghua Zhang

**Affiliations:** Smart Materials and Advanced Structure Laboratory, School of Mechanical Engineering and Mechanics, Ningbo University, Ningbo 315211, China

**Keywords:** composite, multi-walled carbon nanotubes, water absorption, flexible sensor, mechanical performance, strain sensing, hygrothermal aging

## Abstract

Conductive polymers and their composites have been widely applied in different applications, including sensing applications. Herein, we constructed a conductive composite of polypropylene, carbon black, and multi-walled carbon nanotubes (PP/CB/MWCNTs) to experimentally study its sensing behaviors in a humid thermal environment. The as-synthesized PP/CB/MWCNT composite polymer was immersed in simulated sweat in deionized water at 67 °C. Regarding their electrical and mechanical properties, different experimental parameters, such as cyclic loading and hygrothermal aging, were investigated by recording the mass changes, carrying out strain sensing experiments, and performing dynamic mechanical analyses before and after the immersion test. The results reveal that the filler content improved the rate of water absorption but decreased at higher concentrations of the solution. The sensitivity of the material decreased by up to 53% after the hygrothermal ageing and cyclic loading. Moreover, the sensitivity under cyclic compression loading decreased with an increasing immersion time, qualitatively illustrated by an effective quantum tunneling effect and conducting path model. Finally, hygrothermal aging reduced the composite’s glass transition temperature. This reduction was the most significant for specimens immersed in deionized water, ascribed to the moisture absorption, reducing the molecular chain activity.

## 1. Introduction

In recent years, conductive composites have gained widespread attention due to their high sensitivity, wide strain range, and good flexibility. Previous studies suggest that these composites exhibit a different conductive performance, altering their inner networks within the polymer matrix. Several factors influence the internal conductive network, including the strain and temperature and corresponding changes in their resistance; therefore, conductive polymer materials are widely used as strain sensors in motion-monitoring wearable sensors [1,2,3]. Due to their wide range of applications, the effect of extreme environmental factors such as humidity, chemical media, and temperature on the performance of a composite cannot be ignored.

The selection of a suitable filler and the appropriate matrix is vital to construct conductive composites with high performances. Allaoui et al. [4] studied the effect of different levels of filler content in muti-walled carbon nanotubes (MWCNTs) on the electrical and mechanical performance of conductive composites using an epoxy resin as the matrix. It was found that the strain-sensing Young’s modulus and the yield strength of polymer composites with 1.0 wt% and 4.0 wt% MWCNTs significantly increased. In addition, when the MWCNTs’ content increased from 0.5 wt% and 1.0 wt%, the material turned from an insulator to a conductor. Wichmann et al. [5] studied the electrical conductivity of MWCNTs containing an epoxy resin-based conductive composite at 0.1 wt% and 0.3 wt% with a 0.5 wt% carbon black (CB) content. The obtained results revealed that, under tensile load, the composites exhibited a linear relationship between the strain and resistance conductive composites. In addition, the sensitivity was improved when CB was used as a filler, and the linearity was enhanced when MWCNTs were used as the filler. The durability of epoxy resin conductive polymer materials filled with MWCNTs after a low number of loading cycles was investigated by Vertuccio et al. [6]. The results showed that loading significantly affected the material’s resistance and impedance. Ma et al. [7] investigated the electrical conductivity of epoxy resin-based conductive polymer materials using CB and carbon nanotubes (CNTs) as the fillers. The conductivity of the CNTs was shown to be better than the CB, with a low percolation threshold at 0.2 wt% CB and 0.2 wt% CNTs.

When used as strain-sensing elements, conductive composites are inevitably affected by hygrothermal aging. Several factors, including the humidity and temperature, influence composites’ structures in hygrothermal environments. Diffusion is the most common moisture transport mechanism in conductive polymeric materials and water molecules enter the interior of the composites, making them swell. Another mechanism is transportation through voids and microcracks, activated after the damage produced by water. The presence of water molecules creates defects inside the composite, thus breaking the molecular chains [8,9,10]. It has also been reported that functionalized surfaces with cellulose nanocrystals (which also have high aspect ratios like MWCNTs) [11] cause bundles on PP matrixes to break due to these high-interfacial forces [12]. The aging of the composite material is accelerated when the temperature increases, degrading the physical and chemical properties on a macroscopic scale [13]. Therefore, the moisture diffusion model is crucial in the study of the diffusion mechanism. Recently, hygrothermal aging has attracted widespread attention due to its high efficiency. The mechanical properties of carbon-fiber-reinforced composites were studied by Shaohua Ma et al. [14] under thermal and hygrothermal aging conditions. The water absorption curves of the specimens fit well with Fick’s model, and under the hygrothermal aging condition, the glass transition temperature decreased significantly. Hydrothermal aging significantly affects the sensing properties of materials. Tatjana Glaskova-Kuzmina et al. [15] investigated the electrical properties and water absorption capabilities of carbon nanofillers containing epoxy-based basalt-reinforced composites in an aging environment. It was shown that hydrothermal aging improved the mechanical properties and electrical conductivity of the composites. Researchers have also reported the strain-sensing behavior of conductive composites. For example, Siqi Ding et al. [16] investigated the pressure-sensitive properties of nickel-powder-filled silicone-rubber-based conductive composites under cyclic loading conditions and established a pressure-sensitive model. A qualitative analysis was performed by adopting an effective quantum tunneling effect and conducting path model.

The electrical and mechanical properties of conductive composites under hygrothermal aging or load conditions have been extensively studied; however, the composites’ electrical and mechanical properties after the combined action of hygrothermal aging and load have been less widely investigated [17,18]. Polypropylene (PP) is a common engineering material that is widely used in medical devices due to its good compatibility. It is also applied as a piezoresistive strain sensor due to its excellent mechanical properties, easy processing procedure, easy maintenance, and strong corrosion resistance [19]. Herein, we prepared PP composites using MWCNTs and CB as the fillers. We investigated the effects of simulated sweat moisture and hygrothermal aging conditions on the composites’ mechanical properties and electrical behavior. This research serves as a guide to the use of polypropylene-based conductive composites in flexible sensors for application in the field of monitoring in complex environments such as the combined action of hygrothermal aging and cyclic loading.

## 2. Materials and Methods

### 2.1. Materials

PP (Guangdong Maoming Petrochemical Company, Maoming, China): S700; the melt mass flow rate was 14 g/10 min; and this was used as the matrix for the composite. The matrix was filled with CB and CNTs. CB (Dongguan Donghe Plastic Chemical Co., Ltd., Dongguan, China): masterbatch; the melt mass flow rate was 11.6 g/10 min. MWCNTs (Shandong Dazhan Nanomaterials Co., Ltd., Bingzhou, China): masterbatch; with a purity greater than 97%; the length was 3–12 μm; diameter was 12–15 nm.

### 2.2. Preparation of the Test Samples

The CB and MWCNTs masterbatches were placed in a constant temperature drying oven (DHG-9140A, Sopo, Nanjing, China) at 80 °C for 3 h. The masterbatches were then dried in an electronic balance (PR124ZH, OHAUS Instruments, Kenton County, NJ, USA). Then, the filler as well as the matrix were weighed and proportioned with an electronic balance. The prepared masterbatch was poured into a high-speed blender (ZGH-50, Southeast Equipment Company, Nanjing, China) and mixed thoroughly. In order to disperse the CB and MWCNTs evenly in the matrix, the mixture was melted and co-mixed by a twin-screw extruder. The specimens were placed in a drying oven at 50 °C for 6 h. Finally, the specimens were placed into an injection molding machine (UN120SJ, IZP, Shenzhen, China) for injection molding and then cut to the required size. To minimize the effect of the surface roughness on the water absorption kinetics, all the specimens were polished before testing. The volume fractions of the fillers are shown in Table 1. The XRD (Figure A1) and TGA (Figure A2) plots of the original fabricated composites are shown in the Appendix A.

### 2.3. Hydrothermal Aging

Deionized water, a normal concentration of simulated sweat (1SS), and five-times simulated sweat (5SS) were used in the hygrothermal aging tests at 67 °C (SG-4051B Water Bath, Shanghai Shuoguang Electronic Technology Company, Shanghai, China). The specimens were cut into rectangles of 30 mm × 10 mm × 10 mm. The specimens were immersed for 60 days and weighed periodically. In moisture absorption experiments, the specimens were weighed using a PR124ZH balance (OHAUS Instruments, USA). When the difference in the mass change rate was not greater than 0.01%, the water absorption reached an equilibrium. The 1SS was prepared with a mixture of sodium chloride (20 g) (Shanghai Maclean Biochemical Technology Company, Shanghai, China), ammonium chloride (17.5 g) (Hengxing Chemical Reagent Company, Xuzhou, China), urea (5 g) (Hengxing Chemical Reagent Company, China), lactic acids (15 g) (Guangxiang Silong Science company, Silong, China), acetic acid (2.5 g) (Shanghai Een Chemical Technology Company, Shanghai, China), and deionized water (1 L) (Shanghai Maclean Biochemical Technology Company, Shanghai, China).

### 2.4. Strain Sensing Experiment

Figure 1 shows the experimental setup for the strain sensing. Under compression, the electrical resistance was directly measured using a four-probe method (DMM6500, Keithley, Cleveland, OH, USA) in the range of 100 MΩ at room temperature and samples with a resistivity above 100 MΩ were considered nonconductive due to the limitation of the current setup. The compressive testing under displacement control was conducted using a tensile machine AGS-X (Shimadzu, Kyoto, Japan). The samples were cut into cylindrical specimens with diameters of 9.0 mm ± 0.1 mm and thicknesses of 4.0 mm ± 0.1 mm. To reduce the contact resistance, the contact surface needed to be coated with a uniform layer of conductive silver glue (Humiseal, New York, NY, USA). A cyclic compressive test was carried out after 20 days of immersion, and the experiment was repeated three times after another 20 days of immersion. A triangular loading waveform was used to perform a cyclic compression test with a loading duration of 400 cycles with an upper limit. The upper limit and the lower limit of the displacement amplitude was 0.32 mm and 0.16 mm, respectively, at a loading frequency of 1 Hz.

### 2.5. Characterization

The samples were characterized using different techniques to determine their structure and morphology. Fracture sections of the specimens before and after they underwent hygrothermal aging were observed using scanning electron microscopy (SEM, SU5000 type, Hitachi High-Technologies, Kyoto, Japan). For the SEM, the specimens were place in liquid nitrogen, taken out, and quenched quickly. Gold (SBC-12, KYKY, Beijing, China) was sprayed on the fracture, and the filler distribution in the specimens was observed.

For a dynamic mechanical analysis (DMA) using a DMA-8000 (Perkin-Elmer Corporation, Waltham, MA, USA), the specimens were cut into rectangles of 20 mm × 5 mm × 2 mm. The DMA experiments were conducted with the specimens before and after they underwent hygrothermal aging. This test was performed through a single cantilever beam experiment at a vibration frequency of 1 Hz and an amplitude of 0.05 mm. The heating rate was 5 °C/min, and the experimental temperature was −50~150 °C. Plots of the storage modulus and tanδ vs. temperature were recorded, and the $T_{g}$ was obtained from the tanδ peak.

## 3. Results and Discussion

### 3.1. Mass Variation Curve

Initially, the water absorption of the composites was calculated using the following equation:(1)$$M_{t}=\frac{m_{t}-m_{0}}{m_{0}}\times 100\%$$
where $M_{t}$ represents the rate of water absorption, $m_{t}$ indicates the specimen mass at time *t*, and $m_{0}$ is the initial mass of the specimen. The diffusion process facilitates moisture entering the composite, simulated using Fick’s diffusion model [20,21].

During the hygrothermal aging experiment, the specimens were completely immersed in the medium. As fillers are randomly distributed in their matrices, the specimens were considered isotropic materials; therefore, the water absorption was calculated using the three-dimensional Fick’s model:(2)$$M_{t}=M_{\infty }-(M_{\infty }-M_{0})\frac{8}{\pi^{6}}\sum_{k=1}^{\infty }\sum_{n=1}^{\infty }\sum_{m=1}^{\infty }\frac{{\left[1-{\left(-1\right)}^{k}\right]}^{2}{\left[1-{\left(-1\right)}^{n}\right]}^{2}{\left[1-{\left(-1\right)}^{m}\right]}^{2}}{k^{2}n^{2}m^{2}}\exp\left[-\lambda^{2}{}_{k,n,m}Dt\right]$$
where $\lambda^{2}{}_{k,n,m}=\lambda_{k}^{2}+\lambda_{n}^{2}+\lambda_{m}^{2}={\left(\frac{\pi k}{a}\right)}^{2}+{\left(\frac{\pi n}{b}\right)}^{2}+{\left(\frac{\pi m}{l}\right)}^{2}$ *D* indicates the diffusion coefficient of the material, $M_{t}$ depicts the rate of mass change at time *t*, $M_{\infty }$ is the water absorption rate at saturation, $M_{0}$ is the mass water content at the initial state, *a* is the sample length, *b* represents the sample width, and *c* is the sample height. The diffusion coefficient *D* is determined from the slope of the water absorption curve, $M_{t}(\sqrt{t})$ [22,23]:(3)$$D=\frac{\pi h^{2}}{16t}{\left(\frac{M_{t}-M_{0}}{M_{\infty }-M_{0}}\right)}^{2}$$
where *h* is half of the thickness.

The composites’ masses varied with time when immersed in the deionized water, 1SS, and 5SS at 67 °C, as shown in Figure 2. Figure 2a–d shows that with the increase in filler contents, the water absorption rate at saturation and the time taken to reach saturation increased. Table 2 is the water absorption rate at the time of reaching stability. The highest level of water absorption was observed when the samples were immersed in the deionized water. In contrast, the lowest water absorption level was observed in quintupled simulated sweat. During moisture absorption, the water absorption rate in the simulated sweat saturated compared to that in the deionized water; therefore, it was shown that the ions in the simulated sweat hindered the diffusion of water. The osmotic pressure between the water entering the specimen and the outside concentrated medium reduced the water absorption rate. At the same time, the material’s water absorption rate increased with the increase in filler content. It was found that it fit well with Fick’s model. Tatjana Glaskova-Kuzmina reported similar results [17].

### 3.2. Analysis of the Results of Strain-Sensing Experiments

When PP/CB/MWCNTs composites are applied as strain sensors, their sensitivity and sensing stability under the combined effects of a hygrothermal environment and cyclic loading conditions are very important.

In this experiment, the relative resistance change was calculated as follows:(4)$$\Delta R/R_{0}=\frac{R_{t}-R_{0}}{R_{0}}$$
where $R_{0}$ is the initial resistance, and $R_{t}$ depicts the resistance at any time.

The strain sensitivity factor (*GF*) is defined as the ratio of resistance to strain, which is given as follows:(5)$$GF=\frac{\Delta R}{R_{0}\varepsilon }$$
where ε is the strain.

For the 12-1 specimens aged in deionized water, the normal concentration of simulated sweat, and the five-times sweat, the relative resistance–time was calculated for 20, 40, and 60 days (Figure 3). Under the cyclic loading condition, the resistance decreased initially and then remained stable. All of the other samples immersed in different media followed the same pattern. The sensitivity levels of different specimens under the cyclic load and hygrothermal aging conditions are shown in Table 3. The sensitivity decreased with the increase in immersion time.

Figure 4 shows schematic diagrams of the pressure-sensitive mechanism. When the cyclic load was applied to the specimen, changes in the number of conductive pathways were observed, suggesting a close relationship between the relative resistance and the applied cyclic load on the composite material. When external compressive loading was applied on the composite material, the distance between the adjacent carbon black gradually decreased, reducing the potential barriers and the tunneling resistance. Consequently, the surface electrons jumped through the tunneling effect. In addition, when the composite material was subjected to cyclic loading, the conductive path was continuously formed and destroyed [24,25,26].

A mathematical model needed to be established to further investigate the potential strain-sensing mechanism and determine the data’s reliability. The effect of MWCNTs was ignored to facilitate the modeling, and only the variation in the electrical properties of the CB was investigated.

The current density, J, between adjacent particles was determined, and J is expressed using the current density formula for the tunneling effect [16,27,28,29]:(6)$$J=\beta \frac{3\sqrt{2m\lambda }}{2d}{\left(\frac{e}{h}\right)}^{2}V\exp\left(-4\pi d\sqrt{\frac{2m\lambda }{h^{2}}}\right)$$
where β represents the amplification factor,m indicates the mass of the electron, λ depicts the height of the potential barrier between particles, d illustrates the distance between particles, e is the electron charge, *h* is Planck’s constant, and V is the applied voltage.

Combining Ohm’s law, the tunneling resistance of the thin insulating layer in the polypropylene matrix can be derived as follows:(7)$$R_{m}=\frac{2h^{2}d}{3\beta a^{2}e^{2}\sqrt{2m\lambda }}\exp\left(4\pi d\sqrt{\frac{2m\lambda }{h^{2}}}\right)$$
where *a*^2^ is the effective cross-sectional area.

The resistivity of polypropylene is higher than that of CB; therefore, the resistivity of CB can be neglected in the theoretical modeling of pressure-sensitive properties. When there are *M* conducting particles in each conducting path, the equivalent resistance is calculated using the following equation:(8)$$R=\frac{M-1}{N}\frac{2h^{2}d}{3\beta a^{2}e^{2}\sqrt{2m\lambda }}\exp\left(4\pi d\sqrt{\frac{2m\lambda }{h^{2}}}\right)$$

The number of internal effective paths, *N*, and the inter-particle distance, *d*, change under the compressive load, and the number of conducting particles, *M*, remains constant. N(ε) and d(ε) are both functions concerning strain; the total resistance of the composite is calculated using:(9)$$R(\varepsilon )=\frac{\left(M-1\right)}{N(\varepsilon )}\frac{2h^{2}d(\varepsilon )}{3\beta a^{2}e^{2}\sqrt{2m\lambda }}\exp\left(4\pi d(\varepsilon )\sqrt{\frac{2m\lambda }{h^{2}}}\right)$$

Therefore, the number of paths in the conductive materials is calculated using the following equation:(10)$$N(\varepsilon )=\frac{N(0)}{\exp(A\varepsilon +B\varepsilon^{2}+C\varepsilon^{3}+D\varepsilon^{4})}$$

Under compressive loading, the carbon black particle spacing d(ε) is calculated as follows:(11)$$d(\varepsilon )=d(0)\left(1-\frac{\sigma }{E}\right)=d(0)(1-\varepsilon )$$
where σ is the applied stress, and *E* indicates the elastic modulus. Relative resistance can be calculated by:(12)$$R_{r}(\varepsilon )=\frac{R(\varepsilon )}{R(0)}=\frac{R_{i}(\varepsilon )/N(\varepsilon )}{R_{i}(0)/N(0)}=\frac{d(\varepsilon )/N(\varepsilon )}{d(0)/N(0)}\exp\left\{4\pi \sqrt{\frac{2\pi \lambda }{h^{2}}}\left[d(\varepsilon )-d(0)\right]\right\}$$
(13)$$\frac{\Delta R}{R_{0}}=\frac{R(\varepsilon )}{R(0)}-1=\frac{d(\varepsilon )/N(\varepsilon )}{d(0)/N(0)}\exp\left\{\gamma [d(\varepsilon )-d(0)]\right\}-1$$
where $\gamma =\frac{4\pi \sqrt{2m\lambda }}{h}$. Equations (10) and (11) are substituted into Equation (13).

After substitution, we have:(14)$$\Delta R/R_{0}=(1-\varepsilon )\exp(A^{*}\varepsilon +B\varepsilon^{2}+C\varepsilon^{3}+D\varepsilon^{4})-1$$
(15)$$A^{*}=A+\gamma d(0)$$

Taking the logarithm of Equation (14):(16)$$In(\Delta R/R_{0}+1)=In(1-\varepsilon )+A^{*}\varepsilon +B\varepsilon^{2}+C\varepsilon^{3}+D\varepsilon^{4}$$

The change in relative resistance and the strain can be fitted by Equation (16).

Figure 5 shows the fitted curve of the relative resistance vs. the strain. The model fit well with the experimental data, indicating that the decrease in distance between the carbon blacks increased the number of conductive paths under compressive load conditions and decreased the resistance, i.e., the pressure-sensitive property of the material. The tunneling effect was found to be the main mechanism for the increase in the number of conductive pathways in the composites [30,31].

### 3.3. SEM Results

The fracture sections of the original specimens are shown in Figure 6. It can be seen from the figures that the fillers were well dispersed in the PP matrix after mechanical blending. With the increase in CB content, more conductive paths were formed between the CB or the CB and MWCNTs, reducing the composites’ resistivity.

SEM images of samples 12-1 which underwent aging for different times after cyclic loading are shown in Figure 7. One can see from the figures that, after hygrothermal aging and cyclic loading conditions were applied, the interface between the filler and matrix was bonded well. Although the specimens absorbed moisture under the hygrothermal aging environment and the compressive cyclic loading condition, the materials’ internal structure was not significantly damaged. That is, there were no obvious microdamages or microcracks in the composites. Under the compressive load condition, the distance between the CB decreased; thus, reducing the resistivity.

### 3.4. Dynamic Mechanical Properties

The loss factor curves and storage moduli for the specimens which underwent aging for lengths of time are shown in Figure 8. The storage modulus decreased after immersion due to moisture absorption which resulted in a volume expansion, as shown in Figure 8a–d. Due to a greater level of water absorption, the storage moduli of the specimens immersed in deionized water decreased more significantly than those immersed in simulated sweat. Two peaks were observed between −50 °C and 150 °C, as shown in Figure 8e–h. The first peaks moved to the left with the increase in filler contents, indicating that, when the load was applied, the internal loss increased due to the friction between the filler and the matrix, which decreased. The figure shows a leftward shift after immersion for the specimens with the same contents. The change after immersion in deionized water was found to be more obvious, as shown in Figure 8g,h. The figures show that the change, $T_{g}$, was the absorption of water molecules reducing the activity of the molecular chains. This result agrees well with the water absorption experiment of [13,32]. Storage modulus and loss factor curves for the samples with different filler contents in dry/wet states and different aging times are shown in Appendix A (Figure A3).

## 4. Conclusions

In this research, PP/CB/MWCNT composites were prepared and used to test their water absorption under hygrothermal aging in simulated sweat and deionized water at 67 °C. The obtained results reveal that the water absorption rate at saturation increased with the filler contents, and the highest moisture absorption rate was observed in deionized water. A Fick’s model obtained good fitting results.

The resistance firstly decreased at an 8% cyclic loading strain and a 1 Hz period, and then stabilized, suggesting that PP/CB/MWCNT composites have promising potential in strain-sensing applications. The mathematical model of pressure sensitivity was established by combining the tunnel effect and the effective conducting path model. The relationship between the strain and the resistance change rate fit well, validating the proposed model. Therefore, this research provides a basis to evaluate and monitor the sensor accuracy in practical applications.

As observed in the SEM images, CB and MWCNTs were well dispersed in the PP matrix and bonded well with the matrix. After they underwent hygrothermal aging, the storage modulus decreased more significantly in deionized water than in simulated sweat. In the first peak of tanδ, the curve moved to the left, suggesting that it decreased after undergoing hygrothermal aging, confirming that hygrothermal aging lowered the activity of the molecular chains.

Finally, the composites maintained good sensitivity after hygrothermal ageing and cyclic loading, indicating their greater potential for application in the field of monitoring in complex environments. Alternatively, a combination with dynamic computer simulations [33,34] can be used to evaluate the effect of fillers on composite properties.

## Figures and Tables

**Figure 1 polymers-14-05089-f001:**
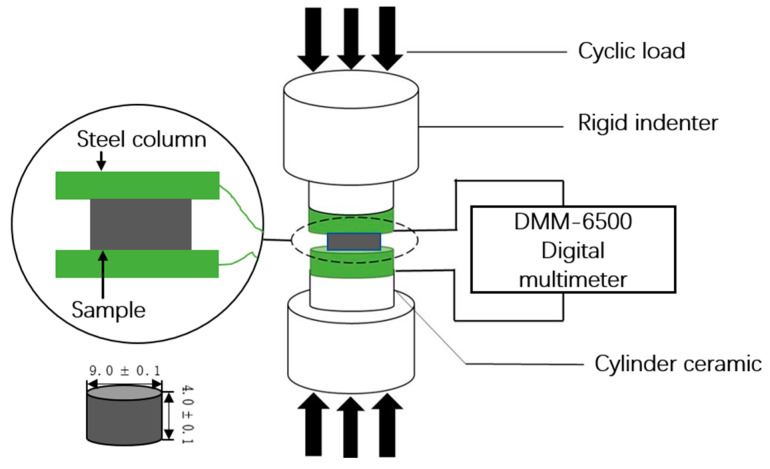
Schematic illustration of the experimental device.

**Figure 2 polymers-14-05089-f002:**
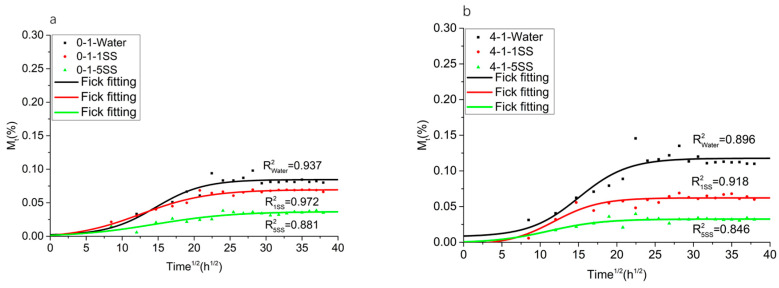
Water uptake of PP/CB/MWCNTs in deionized water and simulated sweat at 67° at different ratios of (**a**) 0-1; (**b**) 4-1; (**c**) 8-1; (**d**) 12-1.

**Figure 3 polymers-14-05089-f003:**
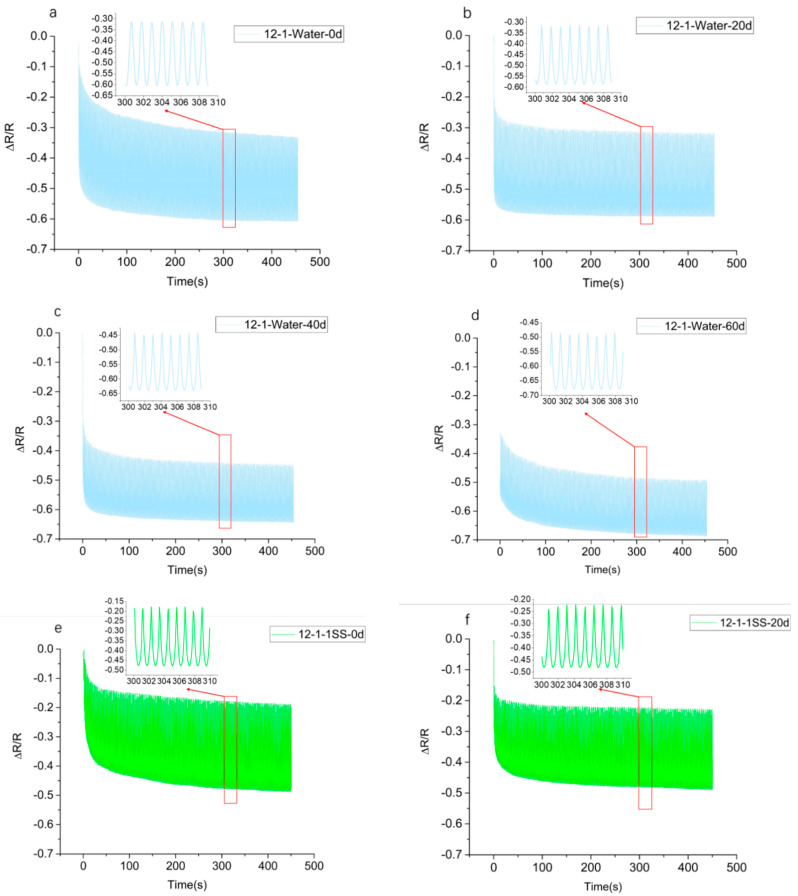
Relative resistance–time relationship curves for 12-1 specimens at 400 cycles of compressive loading after hygrothermal aging immersed in a different solution. (**a**–**d**) The solution is deionized water; (**e**–**h**) the solution is normal concentration of simulated sweat; (**i**–**l**) the solution is 5-times simulated sweat.

**Figure 4 polymers-14-05089-f004:**
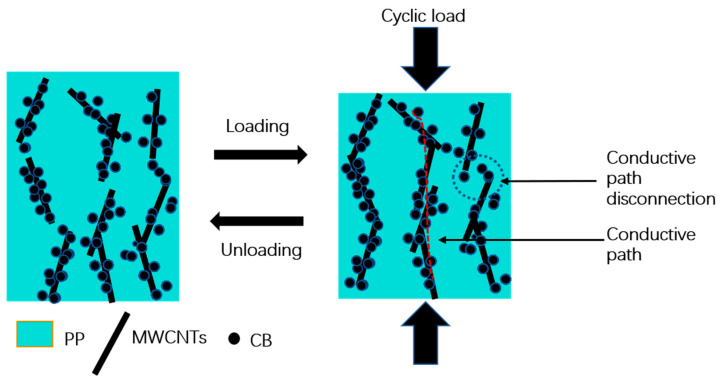
Schematic illustration of pressure-sensitive mechanism.

**Figure 5 polymers-14-05089-f005:**
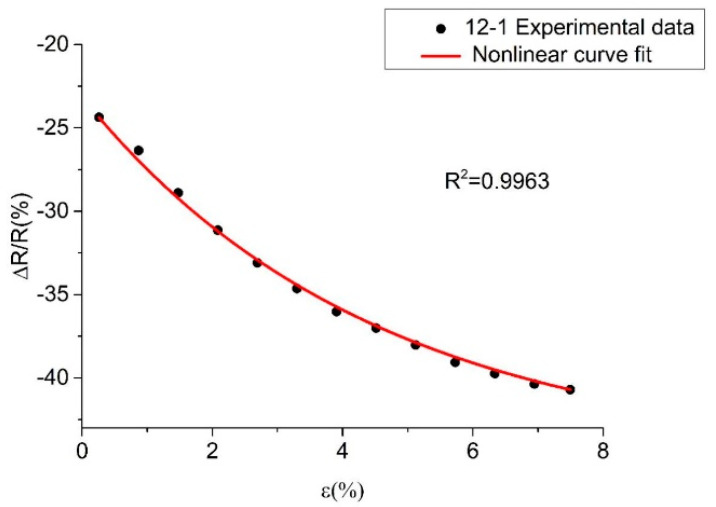
$\frac{∆R}{R}-\varepsilon$ fitting curves of PP/CB/MWCNTs.

**Figure 6 polymers-14-05089-f006:**
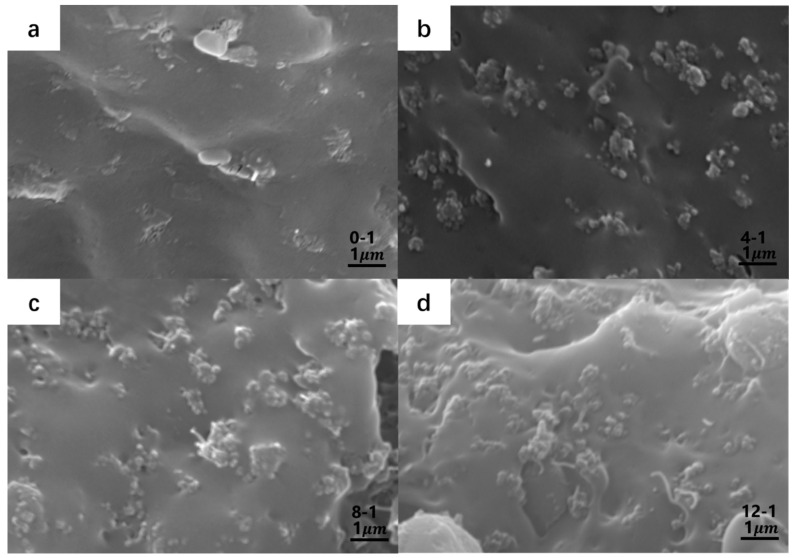
SEM image of (**a**) 0-1; (**b**) 4-1; (**c**) 8-1; (**d**) 12-1.

**Figure 7 polymers-14-05089-f007:**
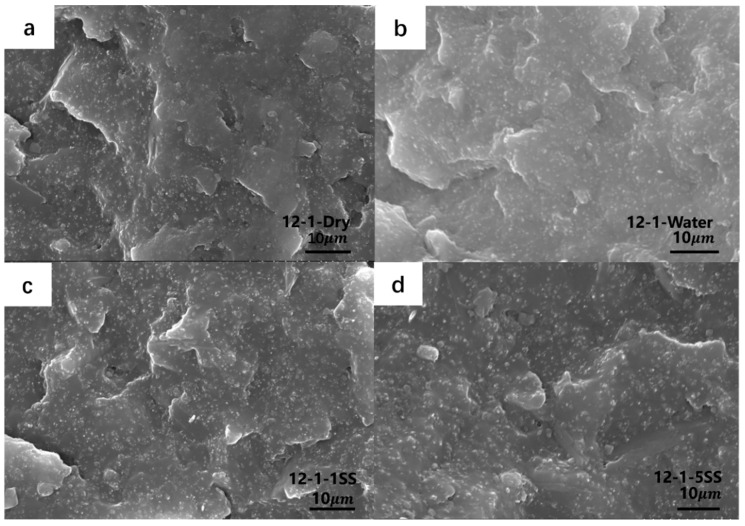
SEM image of 12-1 with different hygrothermal aging time after cyclic loading. (**a**) 12-1-Dry; (**b**) 12-1-Water; (**c**) 12-1-1SS; (**d**) 12-1-5SS.

**Figure 8 polymers-14-05089-f008:**
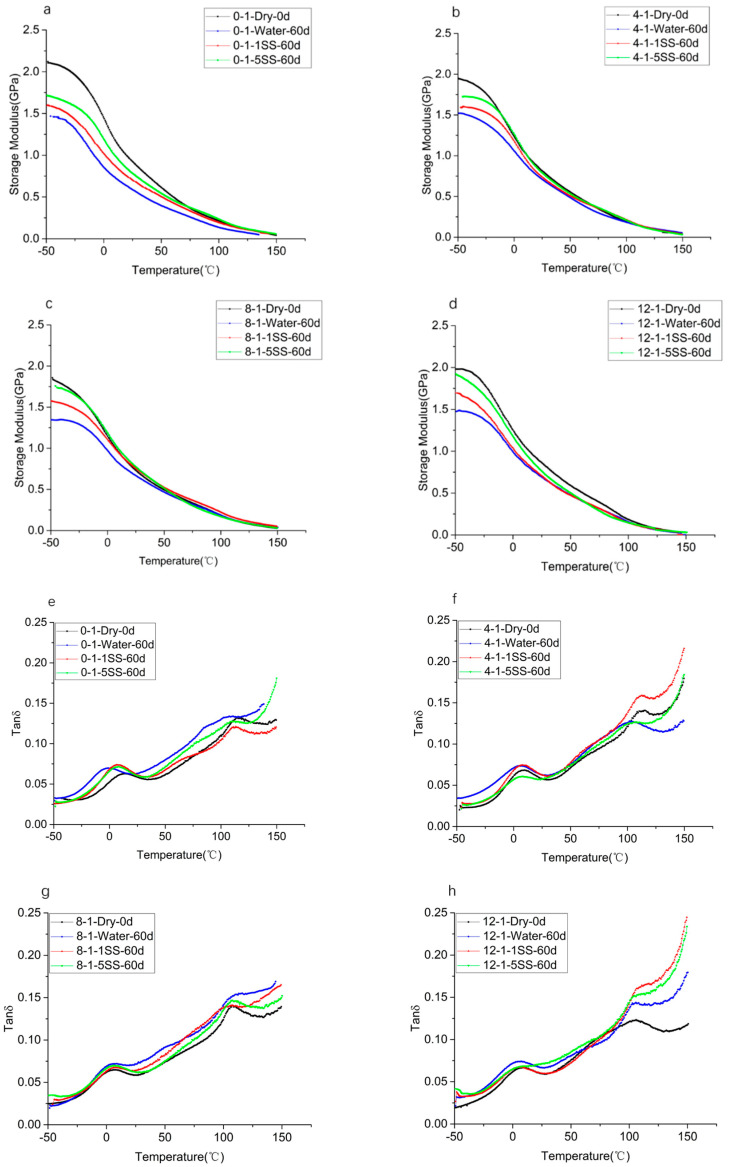
Storage modulus and loss factor curves at different aging times. Storage modulus at (**a**) 0-1; (**b**) 4-1; (**c**) 8-1; (**d**) 12-1; and loss factor at (**e**) 0-1; (**f**) 4-1; (**g**) 8-1; (**h**) 12-1.

**Table 1 polymers-14-05089-t001:** Filler contents of CB/MWCNTs/PP composites.

Content	0-1	4-1	8-1	12-1
CB Vol%	0	4	8	12
MWCNTs Vol%	1	1	1	1

**Table 2 polymers-14-05089-t002:** Water absorption rate in saturated condition.

Solution	0-1	4-1	8-1	12-1
Water	0.082	0.112	0.219	0.238
1SS	0.069	0.064	0.121	0.206
5SS	0.037	0.032	0.101	0.153

**Table 3 polymers-14-05089-t003:** Sensitivity of different specimens before and after aging.

Solution	Dry	20 Days	40 Days	60 Days
Water	3.5	3.3	2.4	2.3
1SS	3.6	3.1	2.3	1.7
5SS	2.5	2.2	1.9	1.6

## Data Availability

The data presented in this study are available on request from the corresponding author.

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
