# Peer review of "Effects of Hygrothermal Aging and Cyclic Compressive Loading on the Mechanical and Electrical Properties of Conductive Composites"

_polymers, 2022, doi:10.3390/polym14235089_

Round 1
Reviewer 1 Report
The manuscript titled “Effects of hygrothermal aging and cyclic compressive loading on the mechanical and electrical properties of conductive polymer composites” by Yi, S.; et al. is an original scientific work where the authors study the electrical and mechanical properties of composites made by polypropylene (PP), carbon black (CB) and multi-walled carbon nanotubes (MWCNTs) though many complementary techniques like strain sensing experiments, moisture absorption tests and scanning electron microscopy (SEM). Four samples with different volume fraction of fillers were tested (0-1, 4-1, 8-1 and 12-1 based on the volume fractions of CB and MWCNTs, respectively). The impact of cyclic compressive loading and hygrothermal aging on the PP/CB/MWCNTs composite mechanical and electrical performance were assessed. The authors found homogeneous dispersion of CB and MWCNTs inner the PP matrix after SEM characterization. The increase of CB/MWCNTs content improves the water absorption at low solution concentrations. Moreover, the glass temperature of PP/CB/MWCNTs is reduced by the hygrothermal aging. The conducted research is highly innovative combining the above described approaches and could have a strong relevance on the composite industry. Moreover, the present methodology can be fully implemented to other matrixes and/or fillers with different nature which could have positive impact on society. The gathered findings may be relevant for the examined field. The results achieved are well-discussed during the main body of the reported manuscript. The scientific paper is well written. In my opinion the present manuscript is innovative and the methodological approached used matches with the scope of Polymers journal. For the above described reasons, I recommend the publication in Polymers once the following remarks will be fixed:
--------
KEYWORDS
The keywords used by the authors are accurate and well-chosen. I may also introduce the following terms “composites” “polypropylene”, “multi-walled carbon nanotubes” and “mechanical performance”.
--------
ABSTRACT
The information provided in the Abstract section is clear and accurate pointing out the most relevant findings of the submitted work. No actions are required for this section.
--------
INTRODUCTION
I) “(…) exhibit variable conductive varying conductive (…)” (line 33). Please, the authors should try to minimize the use of repetitions. For example, this statement could be modified by “(…) exhibit different conductive performance altering their inner networks”.
II) “(…) network changes, including strain and temperature and corresponding changes (…)” (lines 34-35). Please, authors should take care of the verb repetition “change”.
III) “filler content (MWSCNTs)” (line 42). This abbreviation should be defined since it is the first time that it appears in the main body manuscript text.
IV) “(…) 1wt% and 4 wt% (…) 0.5 wt% and 1 wt%, (…) 0.1 wt%, 0.3 wt% and 0.5 wt%” (lines 44-48). Please, authors should homogenize the significant figures.
V) Authors should also enlarge the discussion of the following sentence “Another mechanism is transporting through voids and microcracks, activated after the damage of water” (lines 62-63). Even if I’m agree with this statement, the authors should not discharge that the appearance of breakage phenomena can be also triggered by the interfacial forces exerted between the fillers and the matrix. It has been reported that functionalized surfaces with cellulose nanocrystals (also with high-aspect ratio like MWCNTs) [1] causes the breakage of bundles on PP matrixes due to these high-interfacial forces [2].
VI) “Polypropylene is a common (…). Herein, polypropylene (PP)” (lines 85-88). Please, authors should introduce the abbreviation between brackets just after the first time “Polypropylene” term appears on the text.
[1] Marcuello, C.; et al. Langmuir-Blodgett Procedure to Precisely Control the Coverage of Functionalized AFM Cantilevers for SMFS Measurements: Application with Cellulose Nanocrystals. Langmuir 2018, 34, 9376-9386. https://doi.org/10.1021/acs.langmuir.8b01892.
[2] Berzin, F.; et al. Influence of the polarity of the matrix on the breakage mechanisms of Lignocellulosic fibers during twin-screw extrusion. Polym. Compos. 2020, 41, 1106-1117. https://doi.org/10.1002/pc.25442.
--------
MATERIALS AND METHODS (EXPERIMENTAL WORK)
Authors should add information regarding the country where the products and consumables are coming from.
“Sodium chloride (20 g/L), ammonium chloride (17.5 g/L), urea (5 g/L), and lactic acids (15 g/L). Please, take care of the significant figures.
Finally, authors should provide the population size (N) involved for each type of carried out measurement.
--------
RESULTS AND DISCUSSION
In general terms, the results are displayed clearly for the potential readers. Nevertheless, the following aspects must be further addressed:
I) Figure 2 (line 163). Authors should add the regression coefficients between the experimental data and the Fick model fitting to better compare the correlation between both data. Same comment for Figure 5 (line 242).
II) Figure 6 (line 259). Panels a-d have different magnification respect to e-h which could limit the comprehension by the potential readers. Could it be possible to prepare this Figure with scale bars of 10 µm in all panels and then, plot and inset (e.g. on the upper right side) of zoomed areas of 1 µm scale bar?
III) Figure 7 (line 270). Authors only compare the same condition in dry/wet states and at different aging times. I may strongly encourage the authors to prepare one extra Figure (eventually in Supplementary Information section) where different conditions are merged (e.g. 0-1 Water-60d vs 4-1 Water-60d vs 8-1 Water-60d vs 12-1 Water-60d).
IV) Finally, one table summarizing the quantitative gathered results regarding the properties assessed by the authors should be prepared for the different tested sample conditions.
--------
CONCLUSIONS
This section is clear and concise. Authors well-discussed the main outcomes found in the present manuscript. Authors should introduce some future avenues where their work can be implemented like the potential combination with computer dynamic simulations [3,4] in order to estimate the influence on the performance of composited depending the fillers content embedded in the polymer matrix.
[3] Sun, S.; et al. Effect of Carbon Nanotube Addition on the Interfacial Adhesion between Graphene and Epoxy: A Molecular Dynamics Simulation. Polymers 2019, 11, 121. https://doi.org/10.3390/polym11010121.
[4] Nazarychev, V.M.; et al. Rheological and Mechanical Properties of Thermoplastic Crystallizable Polyimide-Based Nanocomposites Filled with Carbon Nanotubes: Computer Simulations and Experiments. Polymers 2022, 14, 3154. https://doi.org/10.3390/polym14153154.
Finally, authors should state some potential Industrial applications where their research could be fully implemented.
--------
BIBLIOGRAPHY
The reference style is not in the proper format of Polymers journal. Authors should introduce the Journal name in abbreviated form.
--------
OVERVIEW AND FINAL COMMENTS
The submitted work is well-designed and the gathered results are interesting for the design and creation of more durable and sustainable composites with improved mechanical and conductive performance. For this reason, I will recommend the present scientific manuscript for further publication in Polymers once all the aforementioned suggestions will be properly fixed.
Reviewer 2 Report
Current manuscript describes the mechanical and electrical properties of conductive composites based polypropylen. I think it needs major revision before acceptance for publication.
1. Title of manuscript must be changed to:
Effects of hygrothermal aging and cyclic compressive loading on the mechanical and electrical properties of conductive composites.
You can not use the "conductive polymer " term due to PP not being a conductive polymer. Conductive polymers are conjugated polymers such as polyaniline, polypyrrole polythiophene and polyacetylene.
2. Abstract must be enriched with obtained data.
3. Introduction must be improved. Novelty of current work is not clear.
4. Section "2.1 Chemicals and Experimental Methods" is not clear. Please specify it with more details.
5. Figure 1 is not a diagram. Please use the " Schematic illustration"
6. In section "2.4 Characterization" Please add the company and country name of instruments.
7. Please provide the XRD and TGA for fabricated composite.
8.Usually, for measuring the electrical conductivity of composite use four probe technique. What technique did you use for measuring electrical conductivity? Please discuss this in the main text.
Round 2
Reviewer 2 Report
It can be accepted in current form.